# Gut microbiota composition in early pregnancy as a diagnostic tool for gestational diabetes mellitus

Weirong Yao,[1] Ruijing Wen,[2] Zhufeng Huang,[1] Xuhong Huang,[1] Kai Chen,[1] Yuchao Hu,[1] Qianbei Li,[2] Weiqian Zhu,[1] Dejin Ou,[3] Huanlan Bai[2]

**ABSTRACT** Gestational diabetes mellitus (GDM) is a metabolic disorder that poses substantial risks to both maternal and fetal health. Early intervention has been shown to effectively reduce various complications. Gut microbiota dysbiosis is strongly linked to the onset and progression of GDM and may serve as a critical early-warning biomarker. In this study, we systematically analyzed the fecal microbiota of 61 pregnant women during the first trimester using 16S rRNA sequencing. These microbial profiles were correlated with oral glucose tolerance test (OGTT) results at 24–28 weeks of gestation and clinical delivery outcomes. Our analysis identified significant differences in gut microbiota composition between GDM and healthy pregnancies, observed at both the phylum and genus levels early in gestation. Leveraging these microbial distinctions, we developed an early diagnostic model based on genus-level markers, achieving an area under the curve (AUC) of 98.23, indicating high diagnostic precision. This study highlights early-pregnancy microbiota signatures associated with GDM and provides a robust scientific basis for developing microbiota-based diagnostic tools, offering new avenues for GDM prevention and management.

**IMPORTANCE** Gestational diabetes mellitus (GDM) poses significant risks to both maternal and fetal health, but early intervention can reduce complications. This study identifies gut microbiota signatures associated with GDM in the first trimester, providing a potential early diagnostic biomarker. By analyzing fecal microbiota profiles, we developed a diagnostic model with high accuracy (AUC = 98.23). These findings suggest that microbiota-based tools could enable early, non-invasive detection of GDM, offering new opportunities for prevention and personalized management. This research highlights the role of the gut microbiome in pregnancy and has important implications for improving maternal and fetal health outcomes.

**KEYWORDS** gestational diabetes mellitus, gut microbiota, early diagnostic model, biomarkers

G estational diabetes mellitus (GDM) is a prevalent metabolic disorder characterized by abnormal glucose metabolism, primarily manifesting in the mid to late stages of pregnancy (1). GDM significantly elevates the risk of maternal complications such as gestational hypertension, polyhydramnios, and cesarean delivery, while also posing long-term health risks for the fetus, including macrosomia, birth asphyxia, and increased susceptibility to obesity and diabetes in adulthood (2). Current management strategies, predominantly comprising dietary regulation and exercise, are often insufficient to mitigate the adverse maternal and fetal outcomes in patients already presenting with pronounced metabolic disturbances at diagnosis (3). Thus, early prediction and intervention are critical to minimizing the detrimental impacts of GDM.

**Peer Reviewers** Djandan Tadum Arthur Vithran, Central South University, Changsha, China; Choi Hosoon, Central Texas Veterans Healthcare System, Temple, Texas, USA

Address correspondence to Dejin Ou, 2020620649@gzhmu.edu.cn, Huanlan Bai, 1264950200@qq.com, or Weiqian Zhu, 519020408@qq.com.

Weirong Yao, Ruijing Wen, and Zhufeng Huang contributed equally to this article. The author order was determined by descending order of contribution.

The authors declare no conflict of interest.

*[This article was published on 1 July 2025 with incorrect spelling of the author, Dejin Ou. The spelling was corrected in the current version, posted on 17 July 2025.]*

In recent years, researchers have sought to enhance early GDM risk identification through the analysis of serum metabolites, including triglycerides, free fatty acids, branched-chain amino acids (e.g., leucine and isoleucine), and fatty acid derivatives (e.g., acetylcarnitine) (4). Despite these advances, the sensitivity and specificity of these methods remain constrained by considerable inter-individual metabolic variability (5). Similarly, genetic markers such as single-nucleotide polymorphisms in the FTO, TCF7L2, and GCK genes have been explored for risk assessment, but their utility is limited by the inability to fully capture the interactions between genetic predispositions and environmental or lifestyle factors (6). These challenges underscore the urgent need for novel early-warning strategies to address the limitations of existing approaches.

Emerging evidence highlights the pivotal role of the gut microbiota—a complex and dynamic micro-ecosystem—in regulating host metabolism, immune function, and inflammatory responses (7). Dysbiosis of the gut microbiota has been implicated in the pathogenesis of GDM, with significant alterations reported at the phylum and genus levels (8). These include changes in the *Firmicutes*-to-*Bacteroidetes* ratio, increased *Proteobacteria* abundance, reduced beneficial bacteria (e.g., *Bifidobacterium*, *Faecalibacterium*), and elevated pathogenic bacteria (e.g., *Escherichia coli*, *Klebsiella*) (9). Such shifts may impair intestinal barrier function, exacerbate inflammation, and disrupt glucose metabolism (10). These findings suggest that gut microbiota profiling may serve as a promising avenue for early GDM prediction though its characteristics in early pregnancy and clinical applicability require further elucidation. Building on these observations, we hypothesize that first-trimester gut microbiota dysbiosis in early pregnancy precedes clinical diagnosis of GDM and can serve as a clinically actionable reliable predictive biomarker.

In the study, we systematically analyzed the fecal gut microbiota composition of 61 pregnant women at 11–13 weeks of gestation using 16S rRNA sequencing and assessed their oral glucose tolerance test (OGTT) outcomes at 24–28 weeks, along with clinical data at delivery. Our findings reveal significant phylum- and genus-level differences in gut microbiota composition between GDM and healthy pregnant woman (NC) groups in early pregnancy. Leveraging these differences, we developed an early diagnostic model based on genus-level microbial markers, achieving an area under the curve (AUC) of 98.23, indicative of excellent diagnostic performance. This study not only uncovers early-pregnancy gut microbiota features associated with GDM but also provides a scientific basis for the development of microbiota-based diagnostic tools, offering new insights for GDM prevention and management. This study aims to advance the clinical application of gut microbiota for the early prediction of GDM and to identify potential valuable microbial biomarkers for diagnostic models. Furthermore, it provides guidance for subsequent mechanistic exploration of key microbial species.

## RESULTS

### Composition of the gut microbiota in healthy pregnant women during early pregnancy

To investigate the composition of the gut microbiota in healthy pregnant women during early pregnancy, we enrolled 61 healthy women aged 18–40 years (Fig. 1A; Table S1). Participants with type 1 or type 2 diabetes, recent *in vitro* fertilization (IVF) or hormone therapy, recent antibiotic use, or multiple pregnancies were excluded to ensure data reliability. Fecal samples were collected at 11–13 weeks of gestation and systematically analyzed using 16S rRNA sequencing. Blood samples were obtained for OGTT at 24–28 weeks of gestation, and clinical records were collected at delivery (Fig. 1A).

At the phylum level, the predominant components of the gut microbiota at 11–13 weeks were *Firmicutes*, *Bacteroidota*, *Proteobacteria*, *Actinobacteriota*, and *Verrucomicrobiota*, with *Firmicutes* and *Bacteroidota* being dominant and relatively stable across individuals (Fig. 1B). *Firmicutes*, known for their production of short-chain fatty acids (SCFAs), play a critical role in providing energy to intestinal epithelial cells, regulating host metabolism, and modulating immune function, which may be vital for maintaining

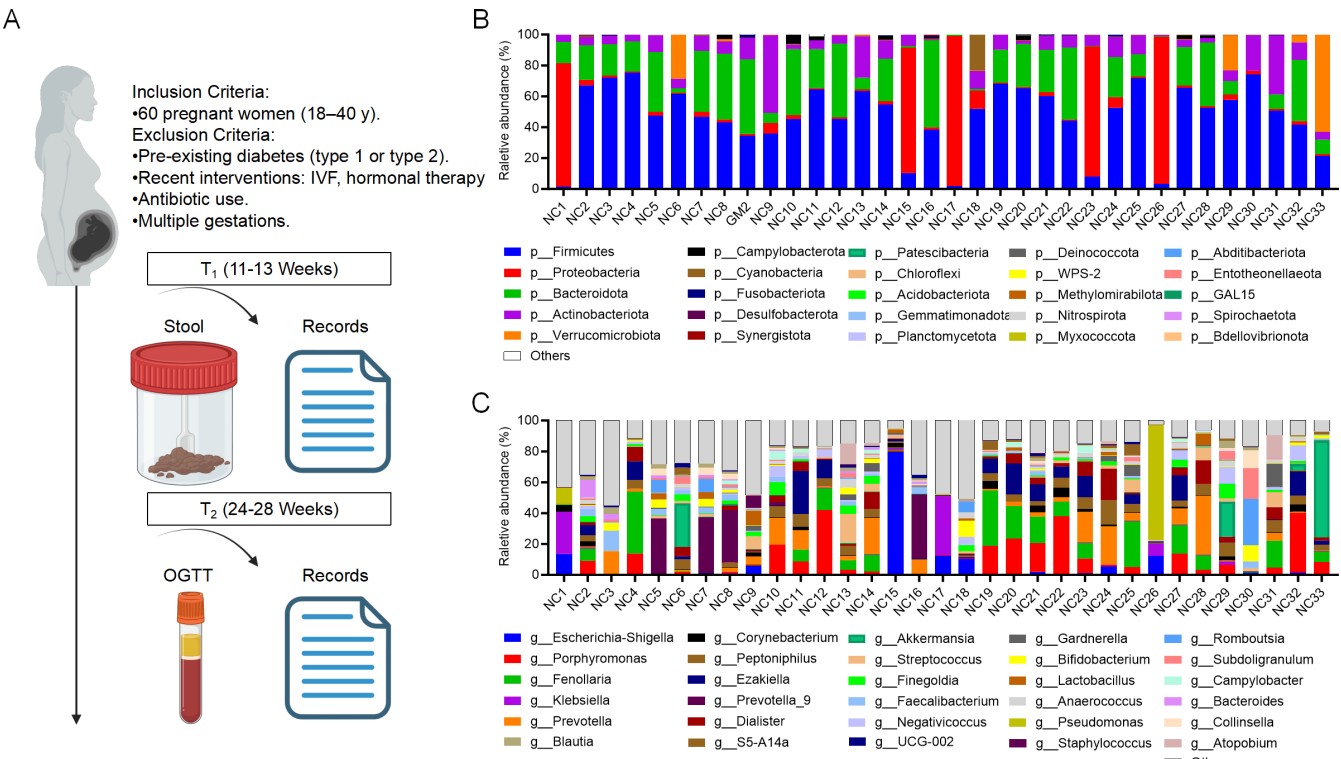

**FIG 1** Microbial composition at phylum and genus levels. (A) Sixty-one pregnant women were prospectively enrolled at 11–13 weeks of gestation. Fecal samples were collected for 16S rDNA sequencing, and clinical data were recorded. OGTT was conducted at 24–28 weeks of gestation. (B and C) Microbial composition profiles at the phylum and genus levels in the control (NC) group.

health during pregnancy (11). *Bacteroidota* are primarily involved in preserving intestinal barrier integrity and regulating inflammatory responses, potentially mitigating pregnancy-associated metabolic disturbances (12). Notably, *Verrucomicrobiota* contribute to maintaining mucosal integrity and immune modulation, which may provide unique protective effects during pregnancy (13).

At the genus level, key taxa identified included *Porphyromonas*, *Escherichia-Shigella*, *Akkermansia*, *Lactobacillus*, *Prevotella*, and *Bifidobacterium* (Fig. 1C). Among these, *Akkermansia* plays a pivotal role in maintaining gut mucosal barrier integrity and immune homeostasis, potentially facilitating a healthy pregnancy through mucosal repair and anti-inflammatory actions (14). *Lactobacillus*, a well-known probiotic genus, supports vaginal microbial balance and prevents pregnancy-associated infections through lactic acid production (15). *Bifidobacterium* contributes to immune modulation and pathogen resistance, further supporting maternal health (16). Additionally, the low abundance of opportunistic pathogens such as *Escherichia-Shigella* may reflect the stability of the gut microbiota during pregnancy, highlighting its balanced nature (17).

Subsequently, we compared the gut microbial composition between the NC and GDM groups. At the phylum level (Fig. S1A), the dominant phyla in both groups were *Firmicutes*, *Bacteroidota*, *Proteobacteria*, and *Actinobacteriota*. Among these, *Firmicutes* exhibited the highest relative abundance, followed by *Proteobacteria* and *Bacteroidota*. Compared with the NC group, the relative abundance of *Firmicutes* was slightly reduced in the GDM group, whereas *Proteobacteria* was increased, suggesting a dysbiosis potentially associated with GDM. Additionally, other phyla such as *Verrucomicrobiota*, *Synergistota*, and *Actinobacteriota* also showed differences between the two groups. At the genus level (Fig. S1B), the microbial composition was more diverse. The predominant genera in both groups included *Escherichia-Shigella*, *Prevotella*, *Blautia*, *Bacteroides*, *Streptococcus*, and *Lactobacillus*. Notably, the GDM group showed an increased relative abundance of potentially pathogenic genera, such as *Escherichia-Shigella* and

*Klebsiella*, while the abundance of beneficial genera including *Bifidobacterium*, *Faecalibacterium*, and *Akkermansia* was markedly reduced. Moreover, taxa categorized as "Others" accounted for a substantial proportion in both groups, indicating a high level of microbial diversity.

These findings underscore the potential roles of these microbial taxa in pregnancy health through mechanisms such as metabolic regulation, immune modulation, and maintenance of gut barrier integrity.

## Analysis of gut microbiota structure and diversity differences between GDM and NC groups during early pregnancy

To investigate differences in the gut microbiota composition and diversity between GDM patients and the NC group at 11–13 weeks of gestation, this study utilized 16S rRNA sequencing combined with various analytical methods. The comparative analysis focused on two aspects: microbial community structure and diversity. Beta diversity was evaluated using principal coordinate analysis (PCoA) and non-metric multidimensional scaling (NMDS). PCoA, based on the weighted UniFrac distance (18), NMDS, based on Bray-Curtis distances emphasizing abundance differences (19), which integrates microbial abundance and evolutionary relationships, revealed only a small separation trend between the GDM and NC groups in two-dimensional space, indicating limited sensitivity of β-diversity metrics to subtle yet biologically meaningful taxonomic shifts (Fig. 2A and B). The limited β-diversity differences between NC and GDM groups may reflect the absence of global microbial restructuring in early-stage GDM. Nevertheless, LEfSe analysis revealed significant alterations in specific taxa, highlighting their potential as early biomarkers. (Fig. 3C and D）

Alpha diversity was assessed using Chao1, Shannon, Simpson, and Pielou_e indices, which, respectively, evaluate species richness, evenness, and community diversity. The

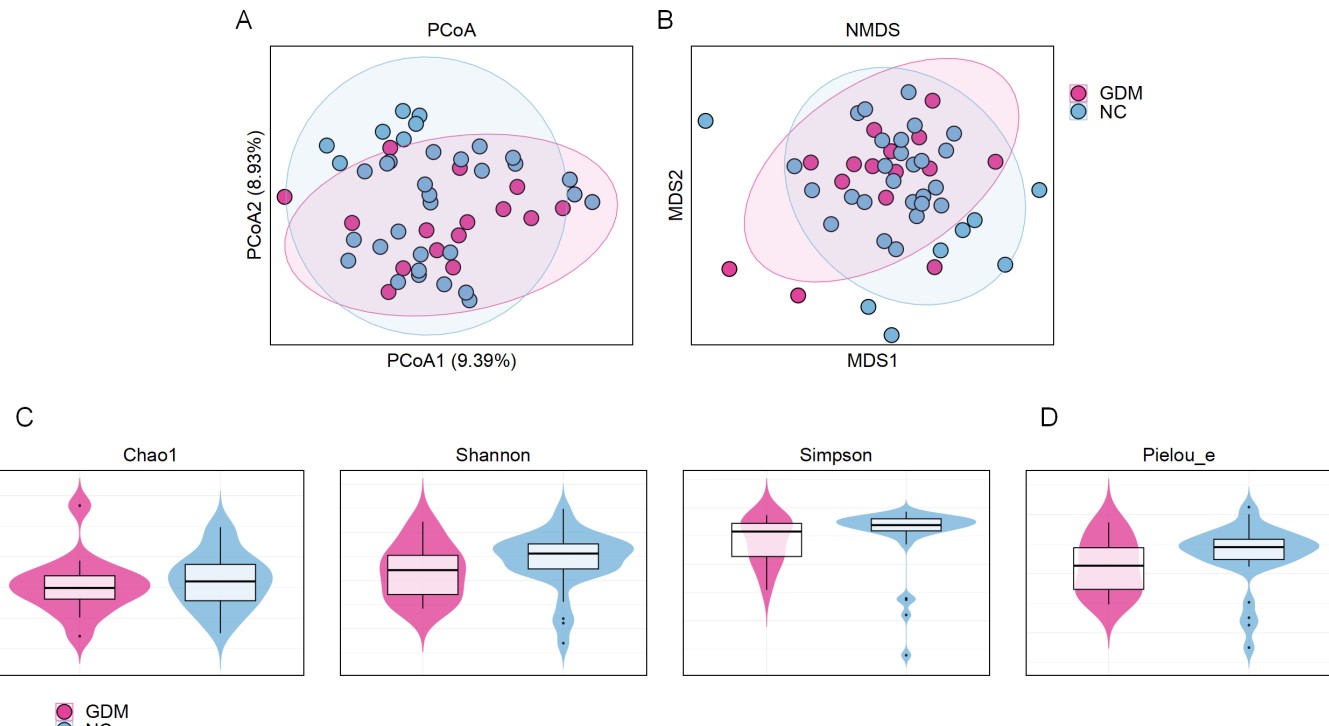

**FIG 2** Microbial diversity and community structure analysis. (A and B) PCoA，based on the weighted UniFrac distance and, NMDS, based on Bray-Curtis distances, which show the fecal microbiota structure in GDM and control (NC) groups. (C) Alpha diversity indices comparing GDM and control groups. Chao1 represents community richness, Shannon and Simpson indices reflect diversity. (D) Pielou's evenness index was used to compare community evenness between the GDM and control groups.

Chao1 index, reflecting the estimated total number of species (20), showed no significant differences in species richness between the GDM and NC groups (Fig. 2C). Both the Shannon and Simpson indices, which account for species richness and evenness (21), indicated similar overall microbial diversity levels in the two groups (Fig. 2C). The Pielou_e index, focusing on species distribution evenness (22), also revealed no significant changes (Fig. 2D). These results suggest that, although the gut microbiota diversity in the GDM group did not exhibit significant reductions, the beta diversity findings indicate that structural abnormalities in the gut microbiota composition had already emerged by 11–13 weeks of gestation.

## Differential analysis of gut microbiota at phylum and genus levels between GDM and NC groups during early pregnancy

A Venn diagram comparison revealed that 19 microbial phyla were shared between the GDM and NC groups at 11–13 weeks of gestation, accounting for 73.08% of the total detected phyla. Additionally, three phyla (11.54%) were unique to the GDM group, while four phyla (15.38%) were unique to the NC group (Fig. 3A). Further analysis indicated that *Proteobacteria* and *Actinobacteriota* were dominant phyla in the GDM group, exhibiting higher relative abundances (Fig. 3C). *Proteobacteria*, often associated with a chronic low-grade inflammatory state, may contribute to GDM pathogenesis. *Firmicutes* play a critical role in host metabolism through the production of SCFAs (23), while *Actinobacteriota* are involved in nutrient absorption and maintaining gut microecological balance (24). In contrast, the NC group showed higher abundances of *Bacteroidota*, which are known for their roles in maintaining intestinal barrier integrity and exerting anti-inflammatory effects (25), potentially providing protective benefits during normal pregnancy.

At the genus level, 430 genera were identified in both groups, representing 59.72% of the total detected genera. Of these, 84 genera (11.67%) were unique to the GDM group, while 206 genera (28.61%) were unique to the NC group (Fig. 3B). The GDM group was enriched with genera such as *Escherichia-Shigella*, *Finegoldia*, and *Klebsiella*, which

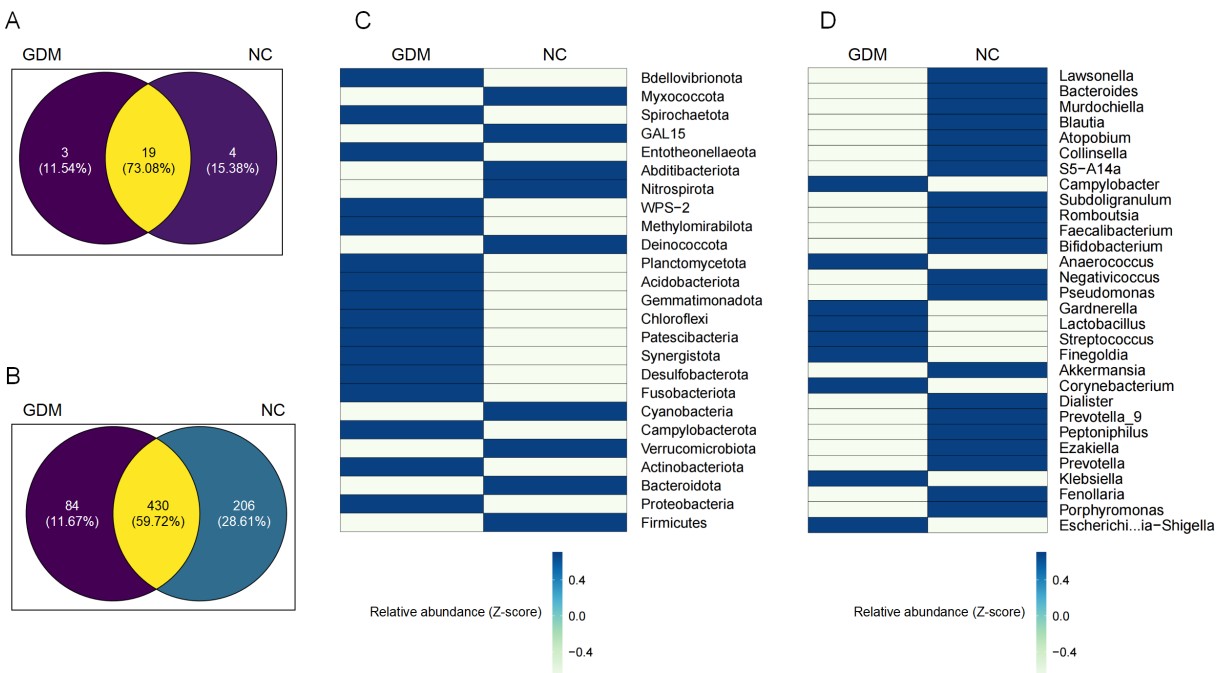

**FIG 3** Unique and shared microbial taxa between GDM and control (NC) groups. Venn diagrams illustrating the unique and shared microbial taxa at the phylum (A) and genus (B) levels in GDM and control groups. Differential microbial taxa identified between GDM and control groups through Lefse (LDA ≥ 3.0, $P < 0.05$) analysis at phylum (C) and genus (D) level. Differential abundance was assessed using the Wilcoxon test ($P < 0.05$). Blue represents relatively enriched microbial communities in GDM, while white indicates relatively depleted microbial communities in GDM.

are often linked to inflammatory responses and metabolic disturbances (Fig. 3D). For instance, *Escherichia-Shigella* is associated with increased intestinal permeability and the release of pro-inflammatory cytokines (26), while *Klebsiella* may contribute to metabolic abnormalities by affecting host insulin sensitivity (27). In contrast, the NC group exhibited higher abundances of genera such as *Bacteroides*, *Prevotella*, and *Faecalibacterium*, which are primarily involved in promoting anti-inflammatory factor production and maintaining mucosal integrity (28). These genera likely support normal metabolic and immune regulation during pregnancy. These findings demonstrate significant alterations in the gut microbiota composition at both phylum and genus levels in GDM patients as early as 11–13 weeks of gestation.

## Functional prediction analysis of gut microbiota in GDM patients

To further explore the functional changes in the gut microbiota of GDM patients at 11–13 weeks of gestation and their potential association with disease mechanisms, we employed functional prediction tools based on 16S rRNA sequencing data. Functional predictions were conducted using PICRUSt2 based on 16S rRNA sequencing data, and the resulting inferred gene content was subsequently annotated with PFAM and TIGRFAM databases. PFAM provides insights into the dynamic changes of specific functional domains, offering key clues on how gut microbiota impact host functions at the molecular level (29). TIGRFAM complements these findings by delivering more precise information on metabolic pathways, protein families, and functional modules, particularly valuable for metabolic network analysis and functional interpretation (30). This multidimensional functional prediction approach ensures comprehensive and robust results.

PFAM analysis revealed that the relative abundance of several functional domains was significantly higher in the GDM group compared to the NC group. Notably enriched domains included the Toxin ToxN family, B12-binding domains, and the Immunity protein 9 domain (Fig. 4A) (31). The Toxin ToxN family, linked to toxin-antitoxin systems, may play a role in stress responses and microbial dysbiosis. Increased B12-binding

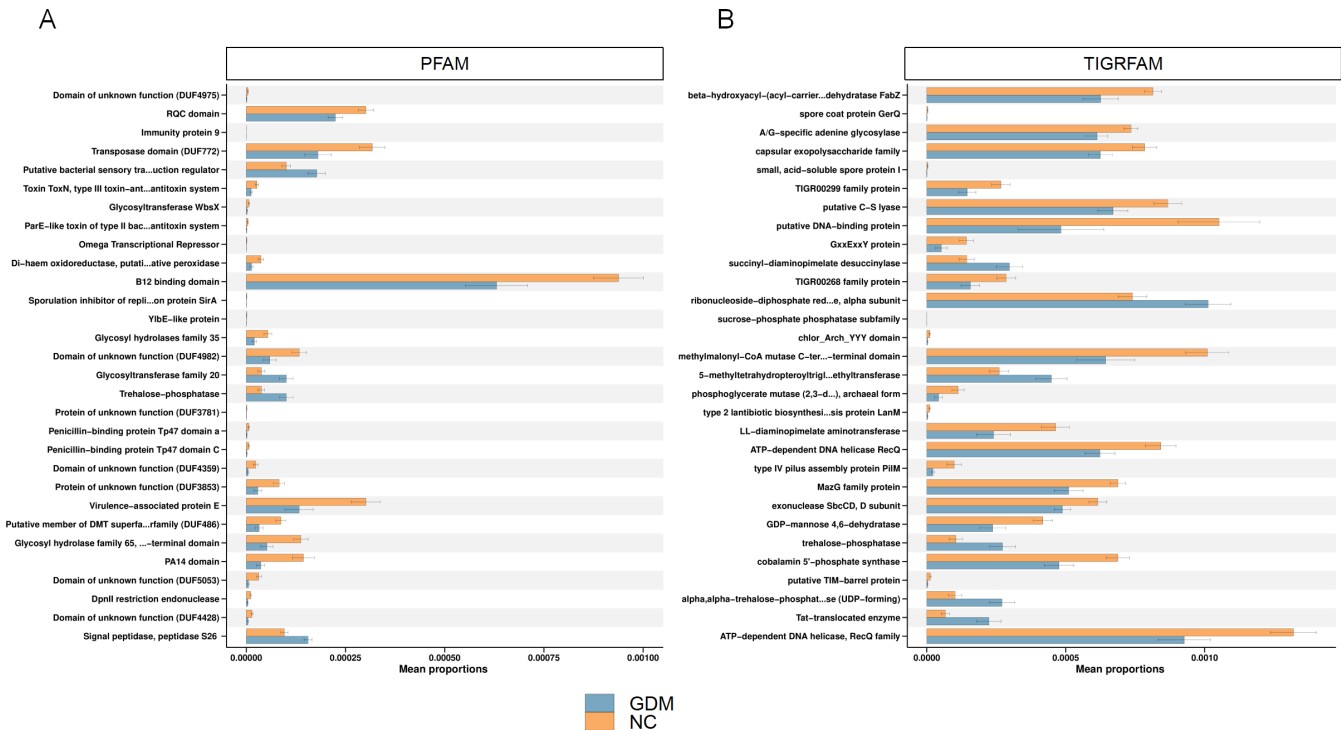

**FIG 4** Pathway enrichment analysis of differential microbial taxa. (A and B) Pathway enrichment analysis of the differential microbiota in GDM and control (NC) groups.

domains suggest potential disruptions in vitamin B12 metabolism, which could indirectly affect host metabolic regulation (32). Changes in the Immunity protein 9 domain might reflect intensified microbial competition and dysregulated host immune responses (33). Additionally, the GDM group exhibited enhanced carbohydrate metabolism functions, such as glycosyl hydrolase family 35 and trehalose-phosphatase, indicating a microbiota-driven inclination toward elevated sugar metabolism, possibly contributing to host metabolic disturbances (34). In contrast, functional domains associated with protein synthesis and transport, including Signal peptidase S26 and Virulence-associated protein E, showed a significant decrease in relative abundance. Reduced Signal peptidase S26 may indicate weakened microbial protein secretion and host-microbiota interactions (35), while the decline in Virulence-associated protein E could suggest reduced microbial competitiveness and exacerbated ecological imbalance (36). A reduction in certain uncharacterized domains, such as DUF3557, may be associated with loss of microbiota adaptability and ecological stability (37).

TIGRFAM analysis revealed similar trends, with significant increases in metabolic and structural functions in the GDM group. Functions associated with metabolic dysregulation, such as beta-hydroxyacyl-ACP-dehydratase (FabZ) and trehalose-phosphatase, were significantly enriched in GDM patients, potentially reflecting abnormal activation of microbial metabolic pathways (Fig. 4B) (38). The GDM group also exhibited enhanced functions related to microbial stability and pathogenicity, including spore coat protein GerQ and type IV pilus assembly protein PilM, which may contribute to microbial dysbiosis and alterations in the host gut barrier (39). Additionally, increased functional activity of the Capsular exopolysaccharide family proteins suggests enhanced microbial adhesion properties, potentially disrupting gut microbiota balance (12). Conversely, several core metabolic functions showed significant reductions in the GDM group. These included cobalamin 5′-phosphate synthase, involved in vitamin B12 biosynthesis, and ATP-dependent DNA helicase RecQ, critical for nucleic acid metabolism and microbial genetic repair. Vitamin B12 plays a pivotal role in host-microbiota co-metabolism, and its decreased synthesis may disrupt host metabolic equilibrium (29). The decline in RecQ helicase activity suggests impaired microbial genetic repair and environmental adaptability. Additionally, reduced activity of sucrose-phosphate phosphatase, related to cell wall stability, may further compromise microbial resilience to external stresses such as inflammation and metabolic disturbances (15). These findings indicate that the functional changes in the gut microbiota of GDM patients exhibit a bidirectional pattern, characterized by enhanced metabolic functions and diminished adaptive capabilities. These alterations may promote the development and progression of GDM by influencing host glucose metabolism, inflammatory responses, and microbial ecological balance.

## Early prediction of GDM based on differential gut microbiota

To evaluate the potential of gut microbiota at 11–13 weeks of gestation for early diagnosis of GDM, this study identified microbial taxa at the phylum and genus levels significantly associated with GDM and developed predictive models to assess their diagnostic performance. At the phylum level, the relative abundances of *Firmicutes* and *Proteobacteria* were significantly higher in the GDM group compared to the NC group, while the abundance of *Bacteroidota* was notably reduced in GDM patients (Fig. 5A). These microbial shifts highlight a significant remodeling of gut microbiota structure in GDM patients, suggesting their potential as diagnostic biomarkers. At the genus level, the GDM group exhibited significant increases in *Escherichia-Shigella* and *Klebsiella*, while *Bacteroides* and *Faecalibacterium* were more abundant in the NC group (Fig. 5B). These alterations at the genus level provide further evidence of gut microbiota dysbiosis in GDM and offer potential microbial markers for diagnosis.

Logistic regression models were constructed using microbial taxa demonstrating significant alterations at both phylum and genus levels, with taxon selection informed by their established biological relevance to glucose metabolism. Diagnostic performance was assessed through receiver operating characteristic (ROC) curve analysis. The

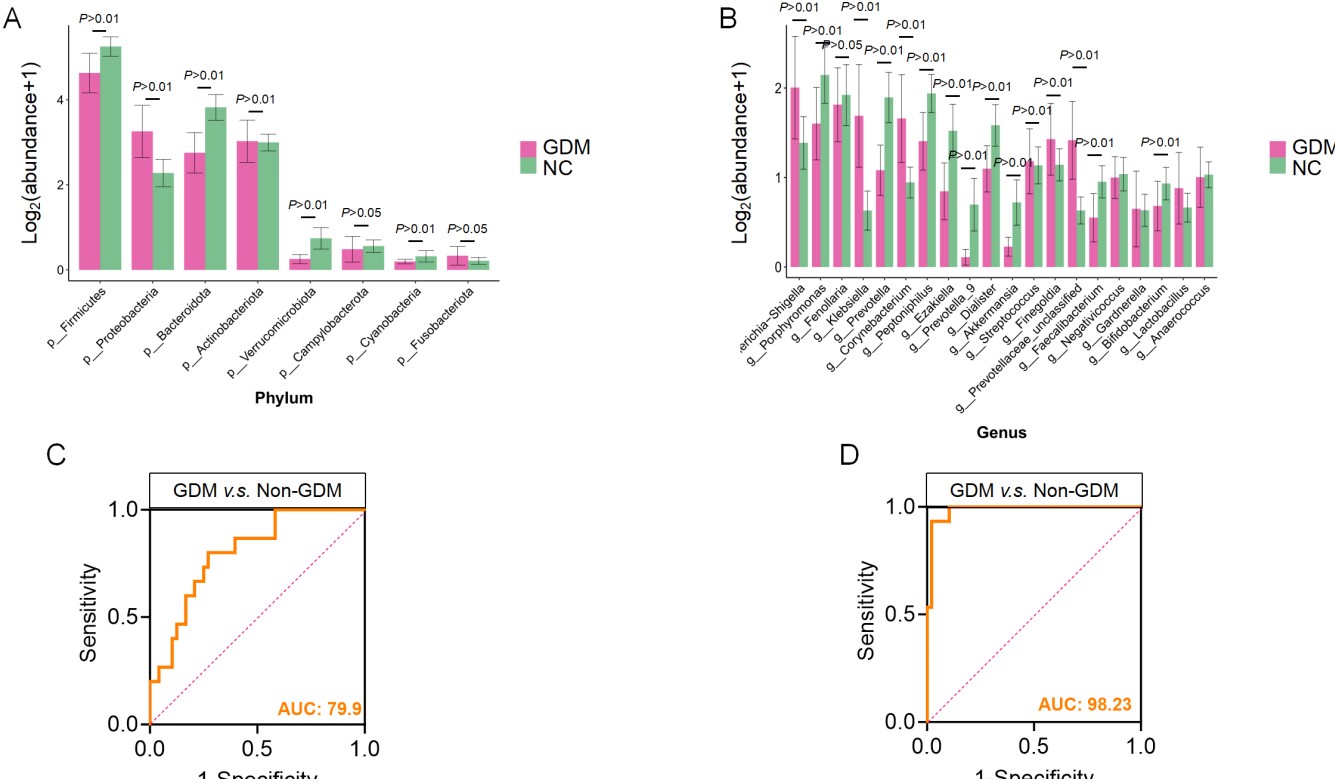

**FIG 5** Differential microbiota abundance and predictive performance. (A and B) Relative abundance of differential microbial taxa at the phylum and genus levels between GDM and control (NC) groups. (C) Receiver operating characteristic (ROC) curve analysis using logistic regression to evaluate the predictive performance of phylum-level taxa for distinguishing GDM from control groups (GDM, $n = 27$; Non-GDM, $n = 34$). (D) ROC curve analysis using logistic regression to evaluate the predictive performance of genus-level taxa for distinguishing GDM from control groups (GDM, $n = 27$; Non-GDM, $n = 34$). AUC (area under the curve): quantifies model performance, where 1.0 indicates perfect prediction and 0.5 represents random chance.

phylum-based diagnostic model (*Firmicutes, Proteobacteria, Bacteroidota, Actinobacteriota, Verrucomicrobiota,* and *Cyanobacteria*) exhibited robust discriminative capacity, yielding an area under the ROC curve (AUC) of 0.799 (Fig. 5C). These findings suggest that phylum-level microbiota signatures may capture structural perturbations of the gut ecosystem associated with GDM pathogenesis, highlighting their potential as screening biomarkers for early disease detection. Notably, the genus-based model (*Porphyromonas, Fenollaria, Klebsiella, Prevotella,* and *Corynebacterium*) achieved exceptional performance with an AUC of 0.982 (Fig. 5D), indicating that fine-scale taxonomic resolution enhances diagnostic precision in this clinical context. Compared to the phylum-level model, genus-level information provided more granular insights into microbiota alterations, effectively capturing features directly linked to GDM pathophysiology. This suggested that genus-level markers are more sensitive and specific for early diagnosis. These findings demonstrate that screening for significantly altered gut microbiota and constructing predictive models can enable precise differentiation of GDM patients from healthy controls. Notably, genus-level microbial changes exhibit remarkable potential for early diagnosis, offering sensitive and specific biomarkers to facilitate timely intervention.

## DISCUSSION

The gut microbiota, an ecosystem comprising trillions of microorganisms, plays a crucial role in maintaining host health by participating in key physiological processes such as metabolism, immune regulation, and intestinal barrier function. During normal pregnancy, the gut microbiota undergoes dynamic changes to meet the host's metabolic and immune demands (22). However, growing evidence indicates that gut dysbiosis

is closely associated with pregnancy-related disorders, including GDM (23). Previous studies suggest that gut microbiota alterations in GDM not only affect microbial composition but may also promote disease progression through inflammatory responses and metabolic dysregulation (29). In this study, we systematically analyzed gut microbiota composition and function in early pregnancy (11–13 weeks) in both GDM patients and healthy controls. Significant differences were observed at both phylum and genus levels, revealing early microbial changes associated with GDM. Using these changes, we developed a high-performance predictive model for early diagnosis, offering a novel approach for GDM identification and intervention.

The absence of significant β-diversity differences aligns with recent findings in early-stage metabolic disorders (40). This phenomenon may arise because incipient dysbiosis primarily affects functionally redundant taxa, leaving overall community structure intact. And our primary aim was to identify predictive microbial signatures for GDM, not to characterize overall community divergence, so even minor taxonomic variations (<1% relative abundance) can serve as robust biomarkers. Previous studies have highlighted significant gut microbiota changes in GDM patients, particularly in mid-to-late pregnancy (31). Alterations in the *Firmicutes*-to-*Bacteroidota* (F/B) ratio, increased *Proteobacteria* abundance, and reduced levels of beneficial taxa such as *Bifidobacteria* have been associated with inflammation and intestinal barrier dysfunction (35). At the genus level, decreases in butyrate-producing bacteria such as *Faecalibacterium* and increases in potential pathogens such as *Escherichia-Shigella* and *Klebsiella* further exacerbate metabolic dysregulation and inflammation (27). Additionally, changes in genera such as *Bacteroides* and *Prevotella* may disrupt SCFA metabolism, affecting glucose homeostasis through the gut-insulin axis (16). Our study extends these findings by showing that significant microbial changes are already present in early pregnancy, prior to clinical diagnosis. Specifically, the GDM group exhibited elevated levels of *Firmicutes* and *Proteobacteria* and reduced levels of *Bacteroidota* at the phylum level. At the genus level, increased abundances of *Escherichia-Shigella* and *Klebsiella* were observed, alongside reduced levels of *Bacteroides* and *Faecalibacterium*. These early microbial alterations provide critical insights into the pathophysiology of GDM and its progression.

The early identification of GDM is critical for mitigating maternal and fetal complications. Current diagnostic methods, such as oral glucose tolerance testing (OGTT) at 24–28 weeks, are limited by their late timing and inability to capture early metabolic abnormalities or microbial changes (18). Emerging approaches, including metabolomics and genomics, have identified potential biomarkers for early GDM detection (20). However, their sensitivity and specificity are constrained by individual variability and their inability to capture the dynamic interactions between host and microbiota (23). Our study addresses these limitations by analyzing gut microbiota at 11–13 weeks of gestation using 16S rRNA sequencing. We identified significant microbial changes and developed predictive models with high diagnostic performance. The phylum-based model demonstrated good discrimination (AUC = 79.9), while the genus-based model achieved exceptional diagnostic accuracy (AUC = 98.23). Compared to metabolomic and genomic methods, our approach offers distinct advantages: 16S rRNA sequencing captures real-time microbial changes associated with GDM; the genus-level model provides fine-grained information directly linked to GDM pathology, offering robust biomarkers for early diagnosis.

Despite these promising findings, several limitations must be acknowledged. First, while our cohort (*n* = 61) achieved sufficient power to detect microbial signatures with moderate effect sizes, future studies with larger samples are warranted to validate generalizability, particularly for low-abundance taxa. And the relatively small sample size (*n* = 61) may limit the generalizability of the results, necessitating validation in larger, multicenter cohorts. Second, functional predictions based on 16S rRNA sequencing lack direct metabolite validation and cannot fully reflect microbial functional activity. Future studies incorporating metabolomic and proteomic analyses are needed to corroborate

these functional insights. Third, while we acknowledge that pre-pregnancy microbiota baselines were not available, all participants were sampled at the same gestational window (8–12 weeks), minimizing temporal variability. Future studies tracking microbiota from pre-conception through pregnancy are needed to establish causality. Lastly, our study identifies microbial associations with GDM but cannot establish causality due to its observational design. Future interventions (e.g., fecal microbiota transplantation in animal models) are needed to test whether microbiota alterations directly contribute to GDM pathogenesis, while this study highlights correlations between gut microbiota changes and GDM, causality remains unproven. Animal models or microbiota transplantation experiments could further elucidate causal mechanisms.

This study systematically revealed significant gut microbiota differences between GDM patients and healthy controls in early pregnancy (11–13 weeks) and developed a highly efficient early diagnostic model. Our findings show that microbial alterations at both phylum and genus levels are present before clinical GDM diagnosis and are closely linked to metabolic dysregulation and inflammation. These results provide a foundation for new strategies in GDM early warning and intervention and lay the groundwork for developing gut microbiota-based diagnostic tools and therapeutic approaches.

In our future research endeavors, we aim to significantly enhance the scope and depth of our study by enrolling a larger cohort and integrating advanced multi-omics approaches, such as fecal metabolomics and plasma proteomics, to achieve a more comprehensive analysis. Our primary objectives include the identification of signature microbial species, a thorough investigation into the potential causal relationships and underlying mechanisms connecting gut microbiota to GDM, and the execution of microbiota transplantation experiments in animal models to validate our findings. This integrated and rigorous approach will not only strengthen the robustness of our conclusions but also provide critical insights into the complex interplay between gut microbiota and GDM, paving the way for novel diagnostic and therapeutic strategies.

## MATERIALS AND METHODS

### Study design and participant recruitment

The study was conducted throughout the pregnancy period with GDM screening performed during the second trimester (T2). Based on screening results, participants were categorized into "developing GDM" and "non-GDM" groups. A total of 65 pregnant women, aged 18–40 years, were recruited from Zhangzhou Second People's Hospital, Fujian Province, between 2023 and 2024, during early pregnancy (11–13 weeks of gestation).

All participants were geographically restricted to residents of Zhangzhou, China, to minimize confounding effects of dietary heterogeneity on gut microbiota composition. This deliberate sampling strategy controlled for regional variations in food consumption patterns, thereby reducing potential bias introduced by inter-individual differences in dietary habits. Exclusion criteria included pre-existing type 1 or type 2 diabetes, exist Functional gastrointestinal disorders, IVF or hormonal treatments in the preceding 3 months, antibiotic use within the last 3 months, and multiple pregnancies. Four participants were lost to follow-up due to relocation or work commitments, resulting in 61 participants completing follow-up through delivery. Clinical data, including GDM diagnoses, were retrieved from electronic medical records. Baseline data, such as height, weight, and fecal samples, were collected at recruitment, while demographic, clinical, and obstetric information, including pregnancy outcomes, was obtained from medical records. Pregnant women who did not experience any diseases or complications that could significantly affect the composition of the gut microbiota throughout the entire pregnancy and had relatively normal biochemical and immune indicators were assigned to the NC group. On the other hand, pregnant women who were diagnosed with GDM during pregnancy and did not have any other conditions that could potentially impact the gut microbiota were assigned to the GDM group.

## GDM diagnosis

GDM was diagnosed using the 75 g OGTT following World Health Organization recommendations. After fasting for at least 8 h, participants underwent venous blood sampling, followed by ingestion of a 75 g anhydrous glucose solution dissolved in 300 mL of water within 5 min. Blood samples were then collected at fasting, 1 h, and 2 h post-ingestion. Diagnostic criteria included fasting glucose ≥ 5.1 mmol/L, 1 h glucose ≥ 10.0 mmol/L, or 2 h glucose ≥ 8.5 mmol/L. GDM was diagnosed if any one criterion was met.

## Fecal sample collection

Participants collected fecal samples at home using tubes containing nucleic acid preservation buffer to prevent degradation of microbial DNA and RNA. Samples were transported to the laboratory within 24 h, aliquoted, and stored at −80°C until analysis.

## DNA extraction and sequencing

Fecal DNA was extracted using the Fecal Genomic DNA Extraction Kit (AU46111-96, BioTeke, China) according to the manufacturer's instructions. DNA concentration was measured using Qubit (Invitrogen, USA). Amplification of the V3–V4 region of the 16S rRNA gene was performed using primers 341F (5′-CCTACGGGNGGCWGCAG-3′) and 805R (5′-GACTACHVGGGTATCTAATCC-3′). The PCR conditions included initial denaturation at 98°C for 10 s, annealing at 54°C for 30 s, extension at 72°C for 45 s, repeated for 32 cycles, and a final extension at 72°C for 10 min. PCR products were purified using AMPure XT Beads (Beckman Coulter, USA), quantified with Qubit, and assessed for quality using an Agilent 2100 Bioanalyzer (Agilent, USA). Sequencing libraries were constructed and sequenced on the Illumina NovaSeq 6000 platform (PE250) by LC-Bio Technologies, Hangzhou, China.

## Sequencing data analysis

Raw sequencing data were processed with cutadapt (v1.9) to remove primers and FLASH (v1.2.8) to merge paired-end reads. Low-quality reads (quality score < 20), sequences shorter than 100 bp, and those with >5% ambiguous bases were filtered using fqtrim (v0.94). High-quality reads were de-duplicated, and chimeric sequences were removed using Vsearch (v2.3.4). Amplicon sequence variants (ASVs) were generated with DADA2. Taxonomy was assigned by aligning sequences to the SILVA and NT-16S databases using QIIME2 plugins. Alpha diversity, beta diversity, and bacterial relative abundance analyses were conducted in QIIME2, and differential abundance was assessed using the Wilcoxon test ($P < 0.05$). Biomarker discovery was performed using LEfSe (LDA ≥ 3.0, $P < 0.05$), and visualizations were created in R (v3.4.4).

## Statistical analysis

The relative abundance of taxonomic features was standardized using $z$-scores transformation to enable cross-sample comparability prior to downstream analyses. Between-group comparisons at phylum and species taxonomic levels were performed using Mann-Whitney $U$ tests with Benjamini-Hochberg false discovery rate (FDR) correction ($q < 0.05$ considered significant). Associations between microbial features and GDM were assessed using Spearman's rank correlation, and linear regression models were applied to adjust for potential confounding factors, such as BMI, age, and gender. A stepwise logistic regression framework was implemented with FDR-significant taxa as predictors and GDM status (NC/GDM) as the dichotomous outcome. Model performance was quantified via receiver operating characteristic (ROC) curve analysis (AUC > 0.7 deemed clinically informative).

## ACKNOWLEDGMENTS

This study was supported by the Zhangzhou Natural Science Foundation Project (ZZ2024J47).

W.Y. was responsible for designing, conducting, and analyzing the experiments. R.W. carried out analyzing the results. H.B. drafted the manuscript, which was reviewed and commented on by all authors. W.Y. provided overall project supervision.

## AUTHOR AFFILIATIONS

[1]The Second Hospital of Zhangzhou, Zhangzhou, China
[2]Nanfang Hospital, Southern Medical University, Guangzhou, China
[3]The Third Affiliated Hospital of Guangzhou Medical University, Guangzhou, China

## AUTHOR ORCIDs

Ruijing Wen  http://orcid.org/0009-0001-7505-8325
Weiqian Zhu  http://orcid.org/0009-0008-7782-9710
Dejin Ou  http://orcid.org/0009-0009-4977-2531
Huanlan Bai  http://orcid.org/0009-0005-1284-5937

## AUTHOR CONTRIBUTIONS

Weirong Yao, Conceptualization, Supervision | Ruijing Wen, Formal analysis | Zhufeng Huang, Writing – review and editing | Xuhong Huang, Writing – review and editing | Kai Chen, Writing – review and editing | Yuchao Hu, Writing – review and editing | Qianbei Li, Writing – review and editing | Weiqian Zhu, Writing – review and editing | Dejin Ou, Writing – review and editing | Huanlan Bai, Writing – original draft

## DATA AVAILABILITY

The data sets generated and analyzed during this study are available from the corresponding author upon reasonable request. The 16S rRNA sequencing data have been deposited in NCBI SRA (accession: PRJNA1254708) .

## ETHICS APPROVAL

This prospective cohort study was approved by the Ethics Committee of Zhangzhou Second People's Hospital (Approval Number: LL2023-21). Participants provided written informed consent before enrollment.

## ADDITIONAL FILES

The following material is available online.

### Supplemental Material

**Figure S1 and Table S1 (Spectrum03390-24-s0001.xml).** Fig. S1: (A and B) Microbial composition profiles at the phylum and genus levels in the NC and GDM group. Table S1: Cohort description.

### Open Peer Review

**PEER REVIEW HISTORY (review-history.pdf).** An accounting of the reviewer comments and feedback.

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
