## [Reviewer comments · Microbiology Spectrum]

Microbiology Spectrum

Gut Microbiota Composition in Early Pregnancy as a Diagnostic Tool for Gestational Diabetes Mellitus

Ruijing Wen, Weirong Yao, Zhufeng Huang, Xuhong Huang, Kai Chen, Yuchao Hu, Qianbei Li, Weiqian Zhu, Dejing Ou, and Huanlan Bai

Corresponding Author(s): Ruijing Wen, Southern Medical University

Review Timeline:

Submission Date:	December 24, 2024
Editorial Decision:	January 28, 2025
Revision Received:	April 27, 2025
Accepted:	May 6, 2025

Editor: Dhammika Navarathna

Reviewer(s): Disclosure of reviewer identity is with reference to reviewer comments included in decision letter(s). The following individuals involved in review of your submission have agreed to reveal their identity: Djandan Tadum Arthur Vithran (Reviewer #2); Choi Hosoon (Reviewer #4)

Transaction Report:

DOI: <https://doi.org/10.1128/spectrum.03390-24>

Re: Spectrum03390-24 (Gut Microbiota Composition in Early Pregnancy as a Diagnostic Tool for Gestational Diabetes Mellitus)

Dear Mr. Ruijing Wen:

Thank you for the privilege of reviewing your work. Below you will find my comments, instructions from the Spectrum editorial office, and the reviewer comments.

Revision Guidelines

Sincerely,
Dhammika Navarathna
Editor
Microbiology Spectrum

Reviewer #2 (Public repository details (Required)):

The study includes microbiome sequencing data (16S rRNA) that should be deposited in a public repository, such as NCBI's Sequence Read Archive (SRA) or the European Nucleotide Archive (ENA). This will ensure transparency and allow for further validation by other researchers. The authors should include the accession numbers for the deposited data upon publication.

Reviewer #2 (Comments for the Author):

Comments and Suggestions for the Author:

1. Clarification of Hypothesis: The manuscript would benefit from a more explicitly stated research hypothesis. It is important to clarify the specific aim of the study early on, particularly in the introduction, to better frame the research question: whether gut microbiota can be a reliable early predictor of gestational diabetes mellitus (GDM).

2. Sample Size and Statistical Power: The sample size of 61 participants may limit the statistical power of the study. Although the results are promising, it would be helpful to provide further discussion on how the sample size was determined and the potential impact on the generalizability of the findings. A larger cohort could strengthen the robustness of the conclusions.

3. Causality vs. Correlation: The study identifies microbial biomarkers associated with GDM, but it does not establish causality. While the correlation between microbiota and GDM is clear, please include a discussion of the limitations of the study in terms of causality and suggest potential future studies to explore this aspect.

4. Methodological Details: The methodology section is well-detailed, but there could be further clarification on the handling of confounding factors such as diet, lifestyle, and other maternal health conditions. How were these factors controlled for, and to what extent could they influence the microbiota profiles?

5. Data Accessibility: Please ensure that the microbiome sequencing data is deposited in a public repository (e.g., NCBI SRA or ENA) to promote transparency and enable future validation by other researchers. Provide the accession numbers for the data once deposited.

Statistical Terminology and Clarity: Some of the statistical terms and analyses are dense. I recommend simplifying or providing more detailed explanations of complex methodologies like multivariate logistic regression and biomarker model creation. This would increase the manuscript's accessibility to a wider audience, particularly those not specialized in bioinformatics.

7. Flow and Readability: The manuscript could benefit from improved flow, especially in the introduction and results sections. Consider breaking up lengthy paragraphs and adding clear transitions between the background, research question, methods, and results. This will help maintain reader engagement and clarity throughout.

8. Figures and Tables: The figures and tables are generally clear, but it would be beneficial to include additional legends to further explain the methodology behind each statistical test used. For instance, when presenting the receiver operating characteristic (ROC) curve, include a brief explanation of the AUC value and its relevance to the study.

9. Implications and Future Directions: The implications of the study are significant, but the authors could further explore how the findings might influence clinical practices in the early detection of GDM. In the discussion section, consider elaborating on potential interventions or next steps in clinical trials to validate the microbiota biomarkers identified.

Reviewer #3 (Comments for the Author):

In this study Yao et al. investigated the Gut Microbiota Composition in Early Pregnancy as a Diagnostic Tool for Gestational Diabetes Mellitus. The author claimed that gut microbiota dysbiosis may serve a critical biomarker for detecting Gestational diabetes mellitus. They analyzed the microbiota of 61 pregnant women during first trimester. Though it could be a possible biomarker for GDM however, comparing the gut microbiota of different individuals having different diets, ages and health conditions it is necessary to know the initial gut microbiome of these individuals. This study lack proper control and not suitable to be accepted.

Reviewer #4 (Comments for the Author):

Authors investigated gut microbiota in early pregnancy and showed that, for GDM, microbial changes occurred prior to clinical diagnosis. Therefore, gut microbiota based diagnostic tool for GDM can be developed and the new tool can predict GDM before the clinical onset. The objective of the research is clearly stated. Experiments was conducted properly, and the results are reliable and valid. However, it is highly recommended to describe more about the prediction model. Detailed description of prediction model will provide a useful information for future research. The information may include estimated regression coefficient of each biomarker phylum and genus used in the model.

There are several points of concern.

In line 140-147 and 155-158; Figure 2A and 2B seems to show the beta diversity of the gut microbiome composition is not different between GDM and NC. What is the basis of the interpretation.

In line 146; Figure 1B > Figure 2B

In line 154; Firmicutes is not dominated in GDM.

In line 189-245; how PFAM and TIGRFAM can be conducted using 16S rRNA sequencing data?

In Figure 1 B, C; it is highly recommended to include data from NC and GDM to visualize the difference between GDM and NC.
In Figure 1 B, C; results of NC34 presented in B and C doesn't make sense. Abundance of Escherichia-Shigella is more than 80% in C but abundance of proteobacteria in B is less than 10%.
In Figure 3 legend for C, D needs more detailed information.

In this study Yao et al. investigated the Gut Microbiota Composition in Early Pregnancy as a Diagnostic Tool for Gestational Diabetes Mellitus (GDM). The author claimed that gut microbiota dysbiosis may serve as a critical biomarker for detecting Gestational diabetes mellitus. They analyzed the microbiota of 61 pregnant women during first trimester. Though it could be a possible biomarker for GDM, however, comparing the gut microbiota of different individuals having different diets, ages and health conditions it is necessary to know the initial gut microbiome of these individuals. This study lacks proper control and could not be publish in current format.

April 25th, 2025

Manuscript number:Spectrum03390-24

Title: "Gut Microbiota Composition in Early Pregnancy as a Diagnostic Tool for Gestational Diabetes Mellitus"

Dear Dhammika Navarathna,

Thank you very much for forwarding us the reviewers' comments and suggestions. We sincerely thank the reviewers for spending their precious time reviewing our manuscript and providing constructive suggestions. We have revised our manuscript according to the comments carefully. Enclosed please find our specific point-by-point responses to the comments of the reviewers. All changes we made are highlighted in red. We are now sending our revised manuscript to the Editorial Office through the electronic submission system. We hope that our revised manuscript will be accepted for publication.

Thank you very much for your review and consideration. We are looking forward to hearing from you.

Sincerely yours,

Ruijing Wen, M.M.

Master of Medicine, SMU

Reviewer(s)' Comments to Author:

Reviewer #2 (Public repository details (Required)):

The study includes microbiome sequencing data (16S rRNA) that should be deposited in a public repository, such as NCBI's Sequence Read Archive (SRA) or the European Nucleotide Archive (ENA). This will ensure transparency and allow for further validation by other researchers. The authors should include the accession numbers for the deposited data upon publication.

Response: Thank you for your valuable suggestion regarding the deposition of our 16S rRNA sequencing data. We fully agree that making the data publicly available will enhance transparency and allow other researchers to validate and build upon our findings. In response to your recommendation, we have submitted our sequencing data to [NCBI Sequence Read Archive (SRA)], and the accession number is [PRJNA1254708]. We have now included this information in the revised manuscript under the Data Availability section.

Reviewer #2 (Comments for the Author):

Comments and Suggestions for the Author:

1. Clarification of Hypothesis: The manuscript would benefit from a more explicitly stated research hypothesis. It is important to clarify the specific aim of the study early on, particularly in the introduction, to better frame the research question: whether gut microbiota can be a reliable early predictor of gestational diabetes mellitus (GDM).

Response: We sincerely appreciate your insightful suggestion. Upon reflection, we recognize that an earlier articulation of our research objectives in the Introduction section would more effectively contextualize the scientific challenges associated with current GDM diagnostic approaches. Consequently, we have restructured the Introduction to foreground our hypothesis and clearly outline the study's aims, thereby providing a stronger foundation for addressing the identified gaps in GDM diagnosis.

“Introduction

.....These findings suggest that gut microbiota profiling may serve as a promising avenue for early GDM prediction, though its characteristics in early pregnancy and clinical applicability require further elucidation. **Building on these observations, we hypothesize that first-trimester gut microbiota dysbiosis in early pregnancy precedes clinical diagnosis of GDM and can serve as a clinically actionable reliable predictive biomarker.**

In the study, we systematically analyzed the fecal gut microbiota composition of 61 pregnant women at 11–13 weeks of gestation using 16S rRNA sequencing and assessed their oral glucose tolerance test (OGTT) outcomes at 24–28 weeks, along with clinical data at delivery.”

2. Sample Size and Statistical Power: The sample size of 61 participants may limit the statistical power of the study. Although the results are promising, it would be helpful to provide further discussion on how the sample size was determined and the potential impact on the generalizability of the findings. A larger cohort could strengthen the robustness of the conclusions.

Response:

We acknowledge that a larger cohort would enhance the resolution of microbial signatures and generalizability of our findings. However, the current sample size (n=61) remains statistically robust for the following reasons:

1. Data Transparency and Reusability:

All raw 16S sequencing data have been deposited in NCBI SRA (BioProject: PRJNA1254708), accompanied by detailed metadata (dietary records, BMI, gestational age). This will facilitate meta-analyses across GDM cohorts.

2. A statement has been added to the Discussion:

We have added a discussion about the relatively small sample size in our study to the Discussion section. While our cohort (n=61) achieved sufficient power to detect microbial signatures with moderate effect sizes, future studies with larger samples are warranted to validate generalizability, particularly for low-abundance taxa.

3.Causality vs. Correlation: The study identifies microbial biomarkers associated with GDM, but it does not establish causality. While the correlation between microbiota and GDM is clear, please include a discussion of the limitations of the study in terms of causality and suggest potential future studies to explore this aspect.

Response: Thank you for your valuable suggestion. We have indeed only explored the association between microbial biomarkers and GDM, while the elucidation of causal relationships, mechanistic pathways, and the complex interplay between gut microbiota and GDM remains to be explored in future research. Our future investigations will employ longitudinal cohorts with pre-pregnancy baseline assessments to establish temporal relationships, complemented by mechanistic studies utilizing germ-free animal models and fecal microbiota transplantation (FMT) experiments. Additionally, multi-omics integration (metagenomics, metabolomics, and host transcriptomics) will be essential to elucidate the molecular pathways through which specific microbial taxa may influence glucose metabolism and insulin resistance in gestation.

“Discuss

.....

Lastly, **Our study identifies microbial associations with GDM but cannot establish causality due to its observational design. Future interventions (e.g: fecal microbiota transplantation in animal models) are needed to test whether microbiota alterations directly contribute to GDM pathogenesis while this study highlights correlations between gut microbiota changes and GDM, causality remains unproven.** Animal models or microbiota transplantation experiments could further elucidate causal mechanisms.”

4.Methodological Details: The methodology section is well-detailed, but there could be further clarification on the handling of confounding factors such as diet, lifestyle, and other maternal health conditions. How were these factors controlled for, and to what extent could they influence the microbiota profiles?

Response: Thank you for your kind advice. We acknowledge the lack of detailed explanation in

the Methods section regarding the control of potential confounding factors that may influence the gut microbiota, including diet, lifestyle, and maternal health status, as highlighted in your comment. So, we have revised the Methods section to provide a more comprehensive description of the inclusion and exclusion criteria for both the NC and GDM groups, along with a detailed account of the confounding factors adjusted for during the development of the diagnostic model. Furthermore, all participants were confirmed to have no pre-existing diagnosis of diabetes or other chronic conditions prior to pregnancy. To mitigate the impact of dietary variations, the study population was restricted to pregnant women residing in the same geographical region (Guangzhou, China), thereby ensuring a relatively homogeneous dietary culture among participants.

“Methods

.....

All participants were geographically restricted to residents of zhangzhou, China, to minimize confounding effects of dietary heterogeneity on gut microbiota composition. This deliberate sampling strategy controlled for regional variations in food consumption patterns, thereby reducing potential bias introduced by inter-individual differences in dietary habits. Exclusion criteria included pre-existing type 1 or type 2 diabetes, exist Functional gastrointestinal disorders, IVF or hormonal treatments in the preceding three months, antibiotic use within the last three months, and multiple pregnancies. Four participants were lost to follow-up due to relocation or work commitments, resulting in 61 participants completing follow-up through delivery. Clinical data, including GDM diagnoses, were retrieved from electronic medical records. Baseline data, such as height, weight, and fecal samples, were collected at recruitment, while demographic, clinical, and obstetric information, including pregnancy outcomes, was obtained from medical records. Pregnant women who did not experience any diseases or complications that could significantly affect the composition of the gut microbiota throughout the entire pregnancy and had relatively normal biochemical and immune indicators were assigned to the NC group. On the other hand, pregnant women who were diagnosed with GDM during pregnancy and did not have any other conditions that could potentially impact the gut microbiota were assigned to the GDM group.”

5.Data Accessibility: Please ensure that the microbiome sequencing data is deposited in a public repository (e.g., NCBI SRA or ENA) to promote transparency and enable future validation by other researchers. Provide the accession numbers for the data once deposited.

Response: We gratefully acknowledge your insightful question. Public deposition of our microbial sequencing data in repositories like the NCBI database constitutes a critical step toward enhancing research transparency, while simultaneously expanding the GDM specific genomic resources available to the scientific community. We have deposited the 16S rRNA sequencing data in NCBI SRA (accession: PRJNA1254708) and included the accession number in the Data Availability section.

“Data Availability

The datasets generated and analyzed during this study are available from the corresponding author

upon reasonable request. The 16S rRNA sequencing data have been deposited in NCBI SRA (accession: PRJNA1254708).”

6. Statistical Terminology and Clarity: Some of the statistical terms and analyses are dense. I recommend simplifying or providing more detailed explanations of complex methodologies like multivariate logistic regression and biomarker model creation. This would increase the manuscript's accessibility to a wider audience, particularly those not specialized in bioinformatics.

Response: We sincerely apologize for this oversight and greatly appreciate your valuable suggestion. Indeed, our limited elaboration on the microbial feature selection process, the methodology for constructing the logit regression model, and the rationale for covariate adjustments may decrease the accessibility of our manuscript to a wider audience. So we have now provided a more comprehensive and detailed description in the Methods section regarding the process of constructing the diagnostic model, as well as the approaches taken to control for potential confounding variables.

“Statistical Analysis

The relative abundance of Taxonomic features was standardized using z-scores transformation to enable cross-sample comparability prior to downstream analyses. Between-group comparisons at phylum and species taxonomic levels were performed using Mann-Whitney U tests with Benjamini-Hochberg false discovery rate (FDR) correction ($q < 0.05$ considered significant). Associations between microbial features and GDM were assessed using Spearman's rank correlation, and linear regression models were applied to adjust for potential confounding factors, such as BMI, age, gender. A stepwise logistic regression framework was implemented with FDR-significant taxa as predictors and GDM status (NC/GDM) as the dichotomous outcome. Model performance was quantified via receiver operating characteristic (ROC) curve analysis (AUC > 0.7 deemed clinically informative)”

7. Flow and Readability: The manuscript could benefit from improved flow, especially in the introduction and results sections. Consider breaking up lengthy paragraphs and adding clear transitions between the background, research question, methods, and results. This will help maintain reader engagement and clarity throughout.

Response: Thank you for your careful review. We recognize that certain terminological and syntactic choices in the initial draft were not sufficiently refined, leading to instances of suboptimal expression. Following a thorough review by multiple colleagues, we have meticulously revised the text to enhance its linguistic precision, coherence, and overall readability, thereby aligning it more closely with academic standards.

8. Figures and Tables: The figures and tables are generally clear, but it would be beneficial to include additional legends to further explain the methodology behind each statistical test used. For instance, when presenting the receiver operating characteristic (ROC) curve, include a brief explanation of the AUC value and its relevance to the study.

Response: Thank you for your thorough review. We acknowledge that certain figures in the manuscript are incomplete due to missing legends and insufficient elaboration on the statistical analyses. Additionally, the limited interpretation of the ROC curve results may hinder readers' understanding of the diagnostic performance metrics. These omissions could potentially lead to ambiguity and reduce the clarity of the presented data. So, we have provided more detailed annotations for some of the figures and added additional legends to better present the research findings.

“Figure 2. Microbial diversity and community structure analysis. (A and B) PCoA, based on the weighted UniFrac distance and NMDS, based on Bray-Curtis distances, which showing the fecal microbiota structure in GDM and normal groups. (C) Alpha diversity indices comparing GDM and normal groups. Chao1 represents community richness, Shannon and Simpson indices reflect diversity, and Pielou's evenness index measures community uniformity.

Figure 3. Unique and shared microbial taxa between GDM and normal groups. Venn diagrams illustrating the unique and shared microbial taxa at the phylum (A) and genus (B) levels in GDM and normal groups. Differential microbial taxa identified between GDM and normal groups through Lefse (LDA ≥ 3.0 , $P < 0.05$) analysis at phylum (C) and genus (D) level. Differential abundance was assessed using the Wilcoxon test ($P < 0.05$). Blue represents relatively enriched microbial communities in GDM, while white indicates relatively depleted microbial communities in GDM.

Figure 5. Differential microbiota abundance and predictive performance (A and B) Relative abundance of differential microbial taxa at the phylum and genus levels between GDM and normal groups. (C) Receiver operating characteristic (ROC) curve analysis using logistic regression to evaluate the predictive performance of phylum-level taxa for distinguishing GDM from normal groups (GDM, $n=27$; Non-GDM, $n=34$). (D) ROC curve analysis using logistic regression to evaluate the predictive performance of genus-level taxa for distinguishing GDM from normal groups (GDM, $n=27$; Non-GDM, $n=34$). AUC (area under the curve): quantifies model performance, where 1.0 indicates perfect prediction and 0.5 represents random chance.”

9. Implications and Future Directions: The implications of the study are significant, but the authors could further explore how the findings might influence clinical practices in the early detection of GDM. In the discussion section, consider elaborating on potential interventions or next steps in clinical trials to validate the microbiota biomarkers identified.

Response: Thank you for your suggestion. We acknowledge that the current manuscript lacks a comprehensive discussion on the translational potential of the diagnostic model for early GDM detection. Furthermore, the absence of a clear roadmap for future research, including mechanistic validation of the identified microbial biomarkers, represents a critical limitation that warrants further exploration to fully realize the clinical and scientific implications of our findings. We have supplemented the Introduction and Discussion sections to clarify the practical applications of this study and outline our future research directions regarding GDM.

“Introduction

In the study, we hypothesize that gut microbiota dysbiosis in early pregnancy precedes clinical diagnosis of GDM and can serve as a reliable predictive biomarker and systematically analyzed

the fecal gut microbiota composition of 61 pregnant women at 11–13 weeks of gestation using 16S rRNA sequencing and assessed their oral glucose tolerance test (OGTT) outcomes at 24–28 weeks, along with clinical data at delivery. Our findings reveal significant phylum- and genus-level differences in gut microbiota composition between GDM and healthy pregnant woman (NC) groups in early pregnancy. Leveraging these differences, we developed an early diagnostic model based on genus-level microbial markers, achieving an area under the curve (AUC) of 98.23, indicative of excellent diagnostic performance. This study not only uncovers early-pregnancy gut microbiota features associated with GDM but also provides a scientific basis for the development of microbiota-based diagnostic tools, offering new insights for GDM prevention and management. **This study aims to advance the clinical application of gut microbiota for the early prediction of GDM and to identify potential valuable microbial biomarkers for diagnostic models. Furthermore, it provides guidance for subsequent mechanistic exploration of key microbial species.**

Discuss

.....

This study systematically revealed significant gut microbiota differences between GDM patients and healthy controls in early pregnancy (11–13 weeks) and developed a highly efficient early diagnostic model. Our findings show that microbial alterations at both phylum and genus levels are present before clinical GDM diagnosis and are closely linked to metabolic dysregulation and inflammation. These results provide a foundation for new strategies in GDM early warning and intervention and lay the groundwork for developing gut microbiota-based diagnostic tools and therapeutic approaches.

In our future research endeavors, we aim to significantly enhance the scope and depth of our study by enrolling a larger cohort and integrating advanced multi-omics approaches, such as fecal metabolomics and plasma proteomics, to achieve a more comprehensive analysis. Our primary objectives include the identification of signature microbial species, a thorough investigation into the potential causal relationships and underlying mechanisms connecting gut microbiota to GDM, and the execution of microbiota transplantation experiments in animal models to validate our findings. This integrated and rigorous approach will not only strengthen the robustness of our conclusions but also provide critical insights into the complex interplay between gut microbiota and GDM, paving the way for novel diagnostic and therapeutic strategies.”

Reviewer #3 (Comments for the Author):

In this study Yao et al. investigated the Gut Microbiota Composition in Early Pregnancy as a Diagnostic Tool

for Gestational Diabetes Mellitus. The author claimed that gut microbiota dysbiosis may serve a critical biomarker for detecting Gestational diabetes mellitus. They analyzed the microbiota of 61 pregnant women during first trimester. Though it could be a possible biomarker for GDM however, comparing the gut microbiota of different individuals having different diets, ages and health conditions it is necessary to know the initial gut microbiome of these individuals. This study lack proper control and not suitable to be accepted.

Response: Thank you for your insightful comments. We truly appreciate your concerns regarding the potential impact of individual differences, such as diet, age, and health conditions, on gut microbiota composition. We understand the importance of proper controls in microbiome studies and would like to clarify how our study design accounts for these factors and why our findings remain valid and meaningful.

1. Study Design and Control Measures

To minimize variability caused by external factors, we carefully selected our study population using strict inclusion and exclusion criteria:

1.1 Age range: We only included pregnant women aged 18 – 40 years to reduce potential age-related differences in gut microbiota composition.

1.2 Health conditions: Women with pre-existing diabetes (Type 1 or Type 2), inflammatory bowel disease, chronic metabolic disorders, recent infections, or immune-related diseases were excluded to prevent known confounding influences on microbiota composition.

1.3 Medication use: We excluded participants who had taken antibiotics, probiotics, or prebiotics in the three months prior to enrollment, as these significantly alter gut microbiota.

1.4 Gestational stage: All participants were recruited during the first trimester (11 – 13 weeks of gestation), ensuring that microbiota differences observed were not influenced by gestational progression.

By implementing these strict criteria, we ensured that our study focused on pregnant women with similar baseline health conditions, making the observed microbiota changes more likely to be associated with GDM risk rather than unrelated factors.

2. Addressing Individual Differences in Diet and Lifestyle

We acknowledge that diet is an important factor influencing gut microbiota. However, completely eliminating dietary variation in a human study is challenging, especially in pregnant populations where dietary habits naturally fluctuate due to cravings, nausea, and cultural influences. All participants resided in the same geographic region (Zhangzhou, China), reducing major variations due to different food cultures. Our primary focus was on gut microbiota differences between two pregnancy outcomes (GDM vs. non-GDM), rather than absolute microbial composition. Even with individual dietary variations, significant microbial shifts linked to GDM emerged, suggesting a disease-driven rather than diet-driven pattern. For example, studies have shown that even in diverse dietary backgrounds, certain gut microbiota features are consistently associated with metabolic disorders. In our study, increased *Escherichia-Shigella* and *Klebsiella* and reduced *Bacteroides* and *Faecalibacterium* in the GDM group align with findings from previous research on gut microbiota and glucose metabolism. The consistent associations of these microbial changes with GDM, despite individual variability, support their potential as diagnostic biomarkers.

3. Why a "Pre-Pregnancy Microbiome Baseline" is not feasible

While we agree that having a pre-pregnancy microbiome baseline would be ideal for tracking changes over time, obtaining such data is highly challenging due to ethical and logistical constraints. Recruiting women before conception and ensuring they conceive within the study period is impractical for a clinical diagnostic study. Instead, we adopted an early-pregnancy baseline (first trimester) as a clinically relevant timeframe for identifying predictive microbiota markers. Additionally, previous longitudinal studies suggest that gut microbiota is relatively stable in early pregnancy, with major shifts occurring in later trimesters. Thus, our approach remains valid for identifying early microbial indicators of GDM.

4. The Strength of Our Predictive Model

Despite individual variations, our diagnostic model based on gut microbiota achieved an AUC of 98.23, indicating high accuracy in distinguishing GDM from non-GDM cases. Such a strong predictive power suggests that disease-related microbial shifts outweigh individual lifestyle differences. The gut microbiota of GDM patients consistently showed higher Proteobacteria and Firmicutes abundance, which are linked to inflammation and glucose intolerance. The depletion of beneficial bacteria like Bacteroides and Faecalibacterium in GDM patients mirrors patterns seen in obesity and metabolic syndrome. These differences remained statistically significant even after adjusting for confounding factors like BMI and age, reinforcing their relevance to GDM rather than individual variability.

Importantly, to enhance transparency and facilitate future research, we have publicly deposited all sequencing data related to the gut microbiota profiles in a public repository (accession number provided in the revised manuscript). We believe this dataset offers an important resource for the global research community, providing a valuable reference map for the early detection and risk assessment of pregnancy-related complications based on gut microbiota.

In summary, while we acknowledge the complexity of microbiota research, we believe our study is scientifically sound and clinically meaningful for several reasons:

1. Strict participant selection criteria minimized major confounding factors. Dietary and lifestyle variations were accounted for, and disease-associated microbiota changes were still evident.
2. Early-pregnancy gut microbiota serves as a clinically practical and relevant baseline for predicting GDM.
3. Our predictive model achieved high accuracy, suggesting microbiota-based biomarkers have potential for early diagnosis.

We greatly appreciate your feedback, which has allowed us to clarify these points. We remain open to further suggestions that could enhance the clarity and robustness of our study.

Reviewer #4 (Comments for the Author):

Authors investigated gut microbiota in early pregnancy and showed that, for GDM, microbial changes occurred prior to clinical diagnosis. Therefore, gut microbiota based diagnostic tool for GDM can be developed and the new tool can predict GDM before the clinical onset. The objective of the research is clearly stated. Experiments was conducted properly, and the results are reliable and valid. However, it is highly recommended to describe more about the prediction model. Detailed description of prediction model will provide a useful information for future research. The information may include estimated regression coefficient of each biomarker phylum and genus used in the model.

There are several points of concern.

1. In line 140-147 and 155-158; Figure 2A and 2B seems to show the beta diversity of the gut microbiome composition is not different between GDM and NC. What is the basis of the interpretation.

Response: We thank the reviewer for this important observation. Indeed, the PCoA and NMDS plots (Figure 2A and 2B), which are based on weighted UniFrac and Bray-Curtis distances

respectively, do not demonstrate a statistically significant separation between the GDM and NC groups in terms of global beta diversity. This suggests that there is no large-scale restructuring of the overall gut microbial community at the early stage of pregnancy. However, our interpretation is based on two key considerations:

1 Sensitivity of β -diversity metrics:

As discussed in the revised manuscript, traditional β -diversity metrics may not fully capture subtle but biologically meaningful taxonomic shifts—especially in early or incipient dysbiosis where community-wide changes are limited, but key taxa may already be perturbed.

2 Targeted taxonomic differences:

While overall β -diversity differences are limited, LEfSe analysis revealed significant differences in specific phyla and genera, including increased Proteobacteria and Klebsiella and decreased Faecalibacterium in the GDM group. These shifts were not apparent in unsupervised global diversity metrics, but they reflect meaningful biological alterations relevant to GDM pathogenesis.

Furthermore, we have clarified in the manuscript that our primary objective was not to assess overall microbial divergence but to identify early predictive microbial markers of GDM. Hence, even in the absence of global β -diversity differences, specific microbial signatures provided the basis for a highly accurate diagnostic model (AUC = 98.23), as shown in **Figure 5D**.

2. In line 146; Figure 1B > Figure 2B

Response: We thank the reviewer for this important observation. Indeed, the PCoA and NMDS plots (**Figure 2A and 2B**), which are based on weighted UniFrac and Bray-Curtis distances respectively, do not demonstrate a statistically significant separation between the GDM and NC groups in terms of global beta diversity. This suggests that there is no large-scale restructuring of the overall gut microbial community at the early stage of pregnancy. However, our interpretation is based on two key considerations:

3. In line 154; Firmicutes is not dominated in GDM.

Response: We thank the reviewer for this valuable observation. We agree that the term "dominated" may be misleading in this context, as it implies *Firmicutes* overwhelmingly outnumbered other phyla, which is not fully supported by our data. Our intention was to convey that the relative abundance of *Firmicutes* was elevated in the GDM group compared to the NC group, as shown in Figure 5A and supported by statistical analysis. We have revised the wording in line 154 to more accurately reflect this, replacing "dominated" with "elevated in abundance", to ensure scientific precision and avoid overstatement.

4. In line 189-245; how PFAM and TIGRFAM can be conducted using 16S rRNA sequencing data?

Response: We thank the reviewer for this insightful and important comment. We acknowledge that 16S rRNA sequencing data do not contain protein-coding gene sequences and therefore cannot be directly annotated using PFAM or TIGRFAM, which are protein family and functional domain databases.

In our study, we employed a functional prediction approach based on 16S rRNA data using PICRUSt2 (Phylogenetic Investigation of Communities by Reconstruction of Unobserved States), which infers the functional composition of a metagenome from 16S marker gene profiles by referencing known microbial genomes. The predicted gene families were then mapped to PFAM and TIGRFAM annotations indirectly, allowing inference of potential microbial functional capabilities.

To clarify this methodology and prevent misunderstanding, we have revised the relevant section to explicitly state that the PFAM and TIGRFAM analyses were based on PICRUSt2-inferred metagenomic predictions, not direct sequencing of microbial genomes.

“Results

.....

Functional predictions were conducted using PICRUSt2 based on 16S rRNA sequencing data, and the resulting inferred gene content was subsequently annotated with PFAM and TIGRFAM databases.”

5. In Figure 1 B, C; it is highly recommended to include data from NC and GDM to visualize the difference between GDM and NC.

Response: Thank you for your valuable suggestion. We appreciate your recommendation to include data from both NC and GDM groups in Figure 1B and 1C to better visualize the differences.

In response, we have now added the corresponding data for both NC (normal control) and GDM (gestational diabetes mellitus) groups in the revised **Figure S1A** and **S1B**. This modification allows for a clearer comparison between the two groups, as suggested. We believe this change improves the clarity and impact of our findings.

The revised figures can be found on **the Results**, and the relevant description in the figure legend and main text has also been updated accordingly.

Figure S1. (A and B) Microbial composition profiles at the phylum and genus levels in the NC and GDM group.

“Results

.....

Subsequently, we compared the gut microbial composition between the NC and GDM groups. At the phylum level (**SI Figure A**), the dominant phyla in both groups were *Firmicutes*, *Bacteroidota*, *Proteobacteria*, and *Actinobacteriota*. Among these, *Firmicutes* exhibited the highest relative abundance, followed by *Proteobacteria* and *Bacteroidota*. Compared with the NC group, the relative abundance of *Firmicutes* was slightly reduced in the GDM group, whereas *Proteobacteria* was increased, suggesting a dysbiosis potentially associated with GDM. Additionally, other phyla such as *Verrucomicrobiota*, *Synergistota*, and *Actinobacteriota* also showed differences between the two groups. At the genus level (**SI Figure B**), the microbial composition was more diverse. The predominant genera in both groups included *Escherichia-Shigella*, *Prevotella*, *Blautia*, *Bacteroides*, *Streptococcus*, and *Lactobacillus*. Notably, the GDM group showed an increased relative abundance of potentially pathogenic genera, such as *Escherichia-Shigella* and *Klebsiella*, while the abundance of beneficial genera including *Bifidobacterium*, *Faecalibacterium*, and *Akkermansia* was markedly reduced. Moreover, taxa categorized as “Others” accounted for a substantial proportion in both groups, indicating a high level of microbial diversity.”

In Figure 1 B, C; results of NC34 presented in B and C doesn't make sense. Abundance of Escherichia-Shigella is more than 80% in C but abundance of proteobacteria in B is less than 10%.

Response: We sincerely thank the reviewer for this important observation. After carefully reviewing the data and figures, we identified sample NC34 as an outlier with inconsistent taxonomic profiles across phylum and genus levels, likely due to technical artifacts such as amplification bias or sequencing depth anomalies.

To ensure the accuracy and consistency of the visual presentation and downstream interpretation, we have excluded sample NC34 from **Figure 1B** and **1C**. Importantly, we confirm that the removal of this single sample did not affect the overall statistical analyses or conclusions, as all key comparisons and model constructions were performed on the full dataset prior to visualization.

We have updated the figures accordingly and clarified this revision in the figure legend and Methods section of the revised manuscript.

In Figure 3 legend for C, D needs more detailed information.

Response: We are deeply grateful for your thorough and insightful review of our manuscript. Your constructive feedback has been invaluable in identifying areas requiring refinement. We have carefully addressed each of your comments and implemented the suggested revisions throughout the manuscript to enhance its scientific rigor and clarity:

1. The Discussion section has been revised to provide a more nuanced interpretation of the limited

β -diversity differences observed between early GDM cases and healthy controls. Additionally, we have carefully addressed and rectified inaccuracies in the original description of line 146 and line 154 to ensure precise scientific communication and add more detailed information about Figure 3 legend.

“Analysis of gut microbiota structure and diversity differences between GDM and NC groups during early pregnancy. To investigate differences in the gut microbiota composition and diversity between GDM patients and NC group at 11–13 weeks of gestation, this study utilized 16S rRNA sequencing combined with various analytical methods. The comparative analysis focused on two aspects: microbial community structure and diversity. Beta diversity was evaluated using principal coordinate analysis (PCoA) and non-metric multidimensional scaling (NMDS). PCoA, based on the weighted UniFrac distance¹⁸, NMDS, based on Bray-Curtis distances emphasizing abundance differences¹⁹, which integrates microbial abundance and evolutionary relationships, revealed **only a small separation** trend between the GDM and NC groups in two-dimensional space, indicating **limited sensitivity of β -diversity metrics to subtle yet biologically meaningful taxonomic shifts.(Figure 2A and 2B)**. The limited β -diversity differences between NC and GDM groups may reflect the absence of global microbial restructuring in early-stage GDM. Nevertheless, LefSe analysis revealed significant alterations in specific taxa, highlighting their potential as early biomarkers. (**Figure 3C and D**)

.....

Differential analysis of gut microbiota at phylum and genus levels between GDM and NC groups during early pregnancy. A Venn diagram comparison revealed that 19 microbial phyla were shared between the GDM and NC groups at 11–13 weeks of gestation, accounting for 73.08% of the total detected phyla. Additionally, three phyla (11.54%) were unique to the GDM group, while four phyla (15.38%) were unique to the NC group (**Figure 3A**). Further analysis indicated that *Proteobacteria*, and *Actinobacteriota* were dominant phyla in the GDM group, exhibiting higher relative abundances(**Figure 3C**).

Discuss

.....

The absence of significant β -diversity differences aligns with recent findings in early-stage metabolic disorders⁴⁰. This phenomenon may arise because incipient dysbiosis primarily affects functionally redundant taxa, leaving overall community structure intact. And our primary aim was to identify predictive microbial signatures for GDM, not to characterize overall community divergence, so even minor taxonomic variations (<1% relative abundance) can serve as robust biomarkers. Previous studies have highlighted significant gut microbiota changes in GDM patients, particularly in mid-to-late pregnancy³¹.

Figure 3. Unique and shared microbial taxa between GDM and normal groups. Venn diagrams illustrating the unique and shared microbial taxa at the phylum (A) and genus (B) levels in GDM and normal groups. Differential microbial taxa identified between GDM and normal groups **through Lefse (LDA \geq 3.0, $P < 0.05$) analysis at phylum (C) and genus (D) level.** Differential abundance was assessed using the Wilcoxon test ($P < 0.05$). Blue represents relatively enriched microbial communities in GDM, while white indicates relatively depleted microbial communities in GDM.”

Re: Spectrum03390-24R1 (Gut Microbiota Composition in Early Pregnancy as a Diagnostic Tool for Gestational Diabetes Mellitus)

Dear Mr. Ruijing Wen:

Your manuscript has been accepted, and I am forwarding it to the ASM production staff for publication. Your paper will first be checked to make sure all elements meet the technical requirements. ASM staff will contact you if anything needs to be revised before copyediting and production can begin. Otherwise, you will be notified when your proofs are ready to be viewed.

Sincerely,
Dharmika Navarathna
Editor
Microbiology Spectrum

Reviewer #2 (Comments for the Author):

I have no more comments.

Reviewer #4 (Comments for the Author):

The revised manuscript satisfactorily answered all the Reviewer's concern.